# SAMPLE-EFFICIENT DEEP REINFORCEMENT LEARNING VIA EPISODIC BACKWARD UPDATE

## ABSTRACT

We propose Episodic Backward Update - a new algorithm to boost the performance of a deep reinforcement learning agent by fast reward propagation. In contrast to the conventional use of replay memory with uniform random sampling, our agent samples a whole episode and successively propagates the value of a state into its previous states. Our computationally efficient recursive algorithm allows sparse and delayed rewards to propagate effectively throughout the sampled episode. We evaluate our algorithm on 2D MNIST Maze Environment and 49 games of the Atari 2600 Environment and show that our agent improves sample efficiency with a competitive computational cost.

## 1 INTRODUCTION

Recently, deep reinforcement learning (RL) has been very successful in many complex environments such as the Arcade Learning Environment (Bellemare et al., 2013) and Go (Silver et al., 2016). Deep Q-Network (DQN) algorithm (Mnih et al., 2015) with the help of experience replay (Lin, 1991 & 1992) enjoys more stable and sample-efficient learning process, so is able to achieve super-human performance on many tasks. Unlike simple online reinforcement learning, the use of experience replay with random sampling breaks down the strong ties between correlated transitions and also allows the transitions to be reused multiple times throughout the training process.

Although DQN has shown impressive performances, it is still impractical in terms of data efficiency. To achieve a human-level performance in the Arcade Learning Environment, DQN requires 200 million frames of experience for training, which is approximately 39 days of game play in real time. Remind that it usually takes no more than a couple of hours for a skilled human player to get used to such games. So we notice that there is still a tremendous amount of gap between the learning process of humans and that of a deep reinforcement learning agent. This problem is even more crucial in environments such as autonomous driving, where we cannot risk many trials and errors due to the high cost of samples.

One of the reasons why the DQN agent suffers from such low sample efficiency could be the sampling method over the replay memory. In many practical problems, the agent observes sparse and delayed reward signals. There are two problems when we sample one-step transitions uniformly at random from the replay memory. First, we have a very low chance of sampling the transitions with rewards for its sparsity. The transitions with rewards should always be updated, otherwise the agent cannot figure out which action maximizes its expected return in such situations. Second, there is no point in updating a one-step transition if the future transitions have not been updated yet. Without the future reward signals propagated, the sampled transition will always be trained to return a zero value.

In this work, we propose Episodic Backward Update (EBU) to come up with solutions for such problems. Our idea originates from a naive human strategy to solve such RL problems. When we observe an event, we scan through our memory and seek for another event that has led to the former one. Such episodic control method is how humans normally recognize the cause and effect relationship (Lengyel & Dayan, 2007). We can take a similar approach to train an RL agent. We can solve the first problem above by sampling transitions in an episodic manner. Then, we can be assured that at least one transition with non-zero reward is being updated. We can solve the second problem by updating transitions in a backward way in which the transitions were made. By then,

we can perform an efficient reward propagation without any meaningless updates. This method faithfully follows the principle of dynamic programing.

We evaluate our update algorithm on 2D MNIST Maze Environment and the Arcade Learning Environment. We observe that our algorithm outperforms other baselines in many of the environments with a notable amount of performance boosts.

## 2  RELATED WORKS

Reinforcement learning deals with environments where an agent can make a sequence of actions and receive corresponding reward signals, such as Markov decision processes (MDPs). At time t, the agent encounters a state $s_t$ and takes an action $a_t \in \mathcal{A}$, observes the next state $s_{t+1}$ and receives reward $r_t \in \mathcal{R}$. The agent's goal is to set up a policy $\pi$ to take a sequence of actions so that the agent can maximize its expected return, which is the expected value of the discounted sum of rewards $\mathbb{E}_\pi[\sum_t \gamma^t r_t]$.

Q-learning (Watkins, 1989) is one of the most widely used methods to solve such RL tasks. The key idea of Q-learning is to estimate the state-action value function $Q(s, a)$, generally called as the Q-function. The state-action value function may be characterized by the Bellman optimality equation

$$Q^*(s_t, a) = \mathbb{E}[r_t + \gamma \max_{a'} Q^*(s_{t+1}, a')]. \tag{1}$$

Given the state-action value function $Q(s, a)$, the agent may perform the best action $a^* = \text{argmax}_a Q(s, a)$ at each time step to maximize the expected return.

There are two major inefficiencies in the traditional Q-learning. First, each experience is used only once to update the Q-network. Secondly, learning from experiences in a chronologically forward order is much more inefficient than learning in a chronologically backward order. Experience replay (Lin, 1991 & 1992) is proposed to overcome these inefficiencies. After observing a transition $(s_t, a_t, r_t, s_{t+1})$, the agent stores the transition into its replay buffer. In order to learn the Q-values, the agent samples transitions from the replay in a backward order.

In practice, the state space is extremely large, so it is impractical to tabularize Q-values of all state-action pairs. Deep Q-Network (Mnih et al., 2015) overcomes this issue by using deep neural networks to approximate the Q-function. Deep Q-Network (DQN) takes a 2D representation of a state $s_t$ as an input. Then the information of the state $s_t$ passes through a number of convolutional neural networks (CNNs) and fully connected networks. Then it finally returns the Q-values of each action $a_t$ at state $s_t$. DQN adopts experience replay to use each transition in multiple updates. Since DQN uses a function approximator, consecutive states output similar Q-values. So when DQN updates transitions in a chronologically backward order, often errors cumulate and degrade the performance. So DQN does not sample transitions in a backward order, but uniformly at random to train the network. This process breaks down the correlations between consecutive transitions and reduces the variance of updates.

There have been a variety of methods proposed to improve the performance of DQN in terms of stability, sample efficiency and runtime. Some methods propose new network architectures. The dueling network architecture (Wang et al., 2015) contains two streams of separate Q-networks to estimate the value functions and the advantage functions. Neural episodic control (Pritzel et al., 2017) and model free episodic control (Blundell et al., 2016) use episodic memory modules to estimate the state-action values.

Some methods tackle the uniform random sampling replay strategy of DQN. Prioritized experience replay (Schaul et al., 2016) assigns non-uniform probability to sample transitions, where greater probability is assigned for transitions with higher temporal difference error.

Inspired by Lin's backward use of replay memory, some methods try to aggregate TD values with Monte-Carlo returns. Q($\lambda$) (Harutyunyan et al., 2016) and Retrace($\lambda$) (Munos et al., 2016) modify the target values to allow the on-policy samples to be used interchangeably for on-policy and off-policy learning, which ensures safe and efficient reward propagation. Count-based exploration method combined with intrinsic motivation (Bellemare et al., 2016) takes a mixture of one-step return and Monte-Carlo return to set up the target value. Optimality Tightening (He et al., 2017) ap-

---

**Algorithm 1** Simple Episodic Backward Update (single episode, tabular)

---

1: Initialize the Q- table $Q \in \mathbb{R}^{\mathcal{S} \times \mathcal{A}}$ with zero matrix.
   $Q(s, a) = 0$ for all state action pairs $(s, a) \in \mathcal{S} \times \mathcal{A}$.
2: Experience an episode $E = \{(s_1, a_1, r_1, s_2), \ldots, (s_T, a_T, r_T, s_{T+1})\}$
3: **for** $t = T$ to 1 **do**
4:     $Q(s_t, a_t) \leftarrow r_t + \gamma \max_{a'} Q(s_{t+1}, a')$
5: **end for**

---

plies constraints on the target using the values of several neighboring transitions. Simply by adding a few penalty terms into the loss, it efficiently propagates reliable values to achieve faster convergence.

Our work lies on the same line of research. Without a single change done on the network structure of the original DQN, we only modify the target value. Instead of using a limited number of consecutive transitions, our method samples a whole episode from the replay memory and propagates values sequentially throughout the entire sampled episode in a backward way. Our novel algorithm effectively reduces the errors generated from the consecutive updates of correlated states by a temporary backward Q-table with a diffusion coefficient.

## 3 EPISODIC BACKWARD UPDATE

We start with a simple motivating toy example to describe the effectiveness of episodic backward update. Then we generalize the idea into deep learning architectures and propose the full algorithm.

### 3.1 MOTIVATION

Let us imagine a simple graph environment with a sparse reward (Figure 1, left). In this example, $s_1$ is the initial state and $s_4$ is the terminal state. A reward of 1 is gained only when the agent moves to the terminal state and a reward of 0 is gained from any other transitions. To make it simple, assume that we only have one episode stored in the experience memory: $(s_1 \rightarrow s_2 \rightarrow s_1 \rightarrow s_3 \rightarrow s_4)$. When sampling transitions uniformly at random as Nature DQN, the important transitions $(s_1 \rightarrow s_3)$ and $(s_3 \rightarrow s_4)$ may not be sampled for updates. Even when those transitions are sampled, there is no guarantee that the update of the transition $(s_3 \rightarrow s_4)$ would be done before the update of $(s_1 \rightarrow s_3)$. So by updating all transitions within the episode in a backward manner, we can speed up the reward propagation, and due to the recursive update, it is also computationally efficient.

We can calculate the probability of learning the optimal path $(s_1 \rightarrow s_3 \rightarrow s_4)$ for the number of sample transitions trained. With the simple episodic backward update stated in Algorithm 1 (which is a special case of Lin's algorithm (Lin, 1991) with recency parameter $\lambda = 0$), the agent can come up with the optimal policy just after 4 updates of Q-values. However, we see that the uniform sampling method requires more than 30 transitions to learn the optimal path (Figure 1, right).

Note that this method is different to the standard n-step Q-learning (Watkins, 1989).

$$Q(s_t, a_t) \leftarrow (1 - \alpha)Q(s_t, a_t) + \alpha(r_t + \gamma r_{t+1} + \ldots + \gamma^{n-1} r_{t+n-1} + \max_a \gamma^n Q(s_{t+n}, a)). \quad (2)$$

In n-step Q-learning, the number of future steps for target generation is fixed as $n$. However, our method takes $T$ future values in consideration, which is the length of the sampled episode. Also, n-step Q-learning takes $\max$ operator at the $n$-th step only, whereas we take $\max$ operator at every iterative backward steps which can propagate high values faster. To avoid exponential decay of the Q-value, we set the learning rate $\alpha = 1$ within the single episode update.

### 3.2 EPISODIC BACKWARD UPDATE ALGORITHM

The fundamental idea of tabular version of backward update algorithm may be applied to its deep version with just a few modifications. We use a function approximator to estimate the Q-values and generate a temporary Q-table $\tilde{Q}$ of the sampled episode for the recursive backward update. The full algorithm is introduced in Algorithm 2. The algorithm is almost the same as that of Nature DQN (Mnih et al., 2015). Our contributions are the episodic sampling method and the recursive backward

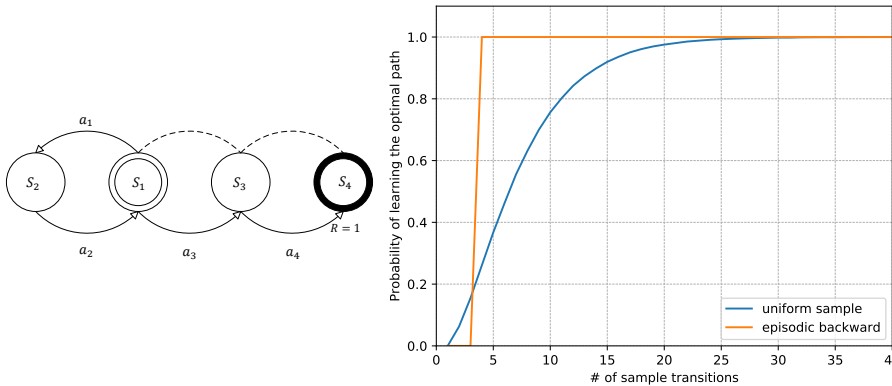

Figure 1: A motivating example for episodic backward update. **Left**: Simple navigation domain with 4 states and a single rewarded transition. **Right**: The probability of learning the optimal path $(s_1 \rightarrow s_3 \rightarrow s_4)$ after updating the Q-values with sample transitions.

---

**Algorithm 2** Episodic Backward Update

---

1:  Initialize replay memory $D$ to capacity $N$
2:  Initialize on-line action-value function $Q$ with random weights $\boldsymbol{\theta}$
3:  Initialize target action-value function $\hat{Q}$ with random weights $\boldsymbol{\theta}^-$
4:  **for** episode = 1 to $M$ **do**
5:      **for** $t = 1$ to Terminal **do**
6:          With probability $\epsilon$ select a random action $a_t$
7:          Otherwise select $a_t = \text{argmax}_a Q(s_t, a; \boldsymbol{\theta})$
8:          Execute action $a_t$, observe reward $r_t$ and next state $s_{t+1}$
9:          Store transition $(s_t, a_t, r_t, s_{t+1})$ in $D$
10:         Sample a random episode $E = \{\boldsymbol{S}, \boldsymbol{A}, \boldsymbol{R}, \boldsymbol{S'}\}$ from $D$, set $T = \text{length}(E)$
11:         Generate temporary target Q table, $\tilde{Q} = \hat{Q}\left(\boldsymbol{S'}, \cdot; \boldsymbol{\theta}^-\right)$
12:         Initialize target vector $\boldsymbol{y} = \text{zeros}(T)$
13:         $\boldsymbol{y}_T \leftarrow \boldsymbol{R}_T$
14:         **for** $k = T - 1$ to 1 **do**
15:             $\tilde{Q}[\boldsymbol{A}_{k+1}, k] \leftarrow \beta \boldsymbol{y}_{k+1} + (1 - \beta)\tilde{Q}[\boldsymbol{A}_{k+1}, k]$
16:             $\boldsymbol{y}_k \leftarrow \boldsymbol{R}_k + \gamma \max_{a \in \mathcal{A}} \tilde{Q}[a, k]$
17:         **end for**
18:         Perform a gradient descent step on $(\boldsymbol{y} - Q(\boldsymbol{S}, \boldsymbol{A}; \boldsymbol{\theta}))^2$ with respect to $\boldsymbol{\theta}$
19:         Every C steps reset $\hat{Q} = Q$
20:     **end for**
21: **end for**

---

target generation with a diffusion coefficient $\beta$ (line number 10 to line number 17 of Algorithm 2), which prevents the errors from correlated states cumulating.

Our algorithm has its novelty starting from the sampling stage. Instead of sampling transitions at uniformly random, we make use of all transitions within the sampled episode $E = \{\boldsymbol{S}, \boldsymbol{A}, \boldsymbol{R}, \boldsymbol{S'}\}$. Let the sampled episode start with a state $S_1$ and contain T transitions. Then $E$ can be denoted as a set of $1 \times n$ vectors, i.e. $\boldsymbol{S} = \{S_1, S_2, \ldots S_T\}$, $\boldsymbol{A} = \{A_1, A_2, \ldots A_T\}$, $\boldsymbol{R} = \{R_1, R_2, \ldots R_T\}$ and $\boldsymbol{S'} = \{S_2, S_3, \ldots S_{T+1}\}$. The temporary target Q-table $\tilde{Q}$, is initialized to store all the target Q-values of $\boldsymbol{S'}$ for all valid actions. $\tilde{Q}$ is an $|\mathcal{A}| \times T$ matrix which stores the target Q-values of all states $\boldsymbol{S'}$ for all valid actions. Therefore the $j$-th column of $\tilde{Q}$ is a column vector that contains $\hat{Q}(S_{j+1}, a; \boldsymbol{\theta}^-)$ for all valid actions $a$ from $j = 1$ to $T$.

Our goal is to estimate the target vector $\boldsymbol{y}$ and train the network to minimize the loss between each $Q(S_j, A_j)$ and $\boldsymbol{y}_j$ for all $j$ from 1 to $T$. After initialization of the temporary Q-table, we perform a recursive backward update. Adopting the backward update idea, one element $\tilde{Q}[\boldsymbol{A}_{k+1}, k]$ in the $k$-th column of the $\tilde{Q}$ is replaced by the next transition's target $y_{k+1}$. Then $y_k$ is estimated using the

maximum of the newly modified $k$-th column of $\tilde{Q}$. Repeating this procedure in a recursive manner until the start of the episode, we can successfully apply the backward update algorithm in a deep Q-network. The process is described in detail with a supplementary diagram in Appendix D.

When $\beta = 1$, the proposed algorithm is identical to the tabular backward algorithm stated in Algorithm 1. But unlike the tabular situation, now we are using a function approximator and updating correlated states in a sequence. As a result, we observe unreliable values with errors being propagated and compounded through recursive max operations. We solve this problem by introducing the diffusion coefficient $\beta$. By setting $\beta \in (0, 1)$, we can take a weighted sum of the newly learnt value and the pre-existing value. This process stabilizes the learning process by exponentially decreasing the error terms and preventing the compounded error from propagating. Note that when $\beta = 0$, the algorithm is identical to episodic one-step DQN.

We prove that episodic backward update with a diffusion coefficient $\beta \in (0, 1)$ defines a contraction operator and converges to optimal Q-function in finite and deterministic MDPs.

**Theorem 1.** *Given a finite, deterministic, and tabular MDP $M = (S, A, P, R)$, the episodic backward update algorithm in Algorithm 2 converges to the optimal Q function w.p. 1 as long as*

- *The step size satisfies the Robbins-Monro condition;*

- *The sample trajectories are finite in lengths $l$: $\mathbb{E}[l] < \infty$;*

- *Every (state, action) pair is visited infinitely often.*

We state the proof of Theorem 1 in Appendix E.

We train the network to minimize the squared-loss between the Q-values of sampled states $Q(\boldsymbol{S}, \boldsymbol{A}; \theta)$ and the backward target $\boldsymbol{y}$. In general, the length of an episode is much longer than the minibatch size. So we divide the loss vector $\boldsymbol{y} - Q(\boldsymbol{S}, \boldsymbol{A}; \theta)$ into segments with size equal to the minibatch size. At each step, the network is trained by a single segment. A new episode is sampled only when all of the loss segments are used for training.

## 4 EXPERIMENTS

Our experiments are designed to verify the following two hypotheses: 1) EBU agent can propagate reward signals fast and efficiently in environments with sparse and delayed reward signals. 2) EBU algorithm is sample-efficient in complex domains and does not suffer from stability issues despite its sequential updates of correlated states. To investigate these hypotheses, we performed experiments on 2D MNIST Maze Environment and on 49 games of the Arcade Learning Environment (Bellemare et al., 2013).

### 4.1 2D MNIST MAZE ENVIRONMENT

We test our algorithm in 2D maze environment with sparse and delayed rewards. Starting from the initial position, the agent has to navigate through the maze to discover the goal position. The agent has 4 valid actions: up, down, left and right. When the agent bumps into a wall, then the agent returns to its previous state. To show effectiveness in complex domains, we use the MNIST dataset (LeCun et al., 1998) for state representation (illustrated in Figure 2). When the agent arrives at each state, it receives the coordinates of the position in two MNIST images as the state representation.

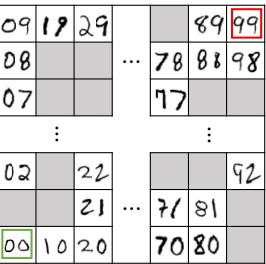

Figure 2: 2D MNIST maze

We compare the performance of EBU to uniform one-step Q-learning and n-step Q-learning. For n-step Q-learning, we set the value of $n$ as the length of the episode. We use 10 by 10 mazes with randomly placed walls. The agent starts at (0,0) and has to reach the goal position at (9,9) as soon as possible. Wall density indicates the probability of having a wall at each position. We assign a reward of 1000 for reaching the goal and a reward of -1 for bumping into a wall. For each wall density, we generate 50 random mazes with different wall locations. We train a total of 50 independent agents, one agent for one maze over 200,000 steps each. The MNIST images for state representation are randomly selected every time the agent visits each state. The relative length is

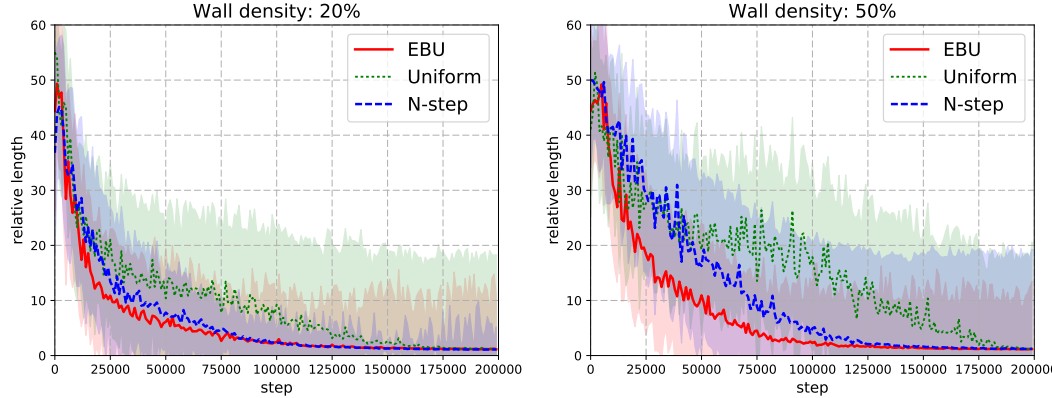

Figure 3: Median relative lengths of EBU and other baseline algorithms. As the wall density increases, EBU outperforms other baselines more significantly. The filled regions indicate the standard deviation of results from 50 random mazes.

Table 1: Relative lengths (Mean & Median) on MNIST Maze after 100,000 steps of training.

| Wall density | EBU (ours) | | Uniform | | N-step | |
|---|---|---|---|---|---|---|
| 20% | 5.44 | 2.42 | 14.40 | 9.25 | **3.26** | **2.24** |
| 30% | **8.14** | **3.03** | 25.63 | 21.03 | 8.88 | 3.32 |
| 40% | **8.61** | **2.52** | 25.45 | 22.71 | 8.96 | 3.50 |
| 50% | **5.51** | **2.27** | 22.36 | 17.90 | 11.32 | 4.93 |

defined as $l_{rel} = l_{agent}/l_{oracle}$, which is the ratio between the length of the agent's path $l_{agent}$ and the length of the ground truth shortest path $l_{oracle}$. Figure 3 shows the median relative lengths of 50 agents over 200,000 training steps. Since all three algorithms achieve median relative lengths of 1 at the end of training, we report the mean and the median relative lengths at 100,000 steps in Table 1. For this example, we set the diffusion coefficient $\beta = 1$. The details of hyperparameters and the network structure are described in Appendix C.

The result shows that EBU agent outperforms other baselines in most of the situations. Uniform sampling DQN shows the worst performance in all configurations, implying the inefficiency of uniform sampling update in environments with sparse and delayed rewards. As the wall density increases, valid paths to the goal become more complicated. In other words, the oracle length $l_{oracle}$ increases, so it is important for the agent to make correct decisions at bottleneck positions. N-step Q-learning shows the best performance with a low wall density, but as the wall density increases, EBU shows better performance than n-step Q. Especially when the wall density is 50%, EBU finds paths twice shorter than those of n-step Q. This performance gap originates from the difference between the target generation methods of the two algorithms. EBU performs recursive max operators at each positions, so the optimal Q-values at bottlenecks are learned faster.

### 4.2 ARCADE LEARNING ENVIRONMENT

The Arcade Learning Environment (Bellemare et al., 2013) is one of the most popular RL benchmarks for its diverse set of challenging tasks. The agent receives high-dimensional raw observations from exponentially large state space. Even more, observations and objectives of the games are completely different over different games, so the strategies to achieve high score should also vary from game to game. Therefore it is very hard to create a robust agent with a single set of networks and parameters that can learn to play all games. We use the same set of 49 Atari 2600 games which was evaluated in Nature DQN paper (Mnih et al., 2015).

We compare our algorithm to four baselines: Nature DQN, Optimality Tightening (He et al., 2017), Prioritized Experience Replay (Schaul et al., 2016) and Retrace($\lambda$) (Munos et al., 2016). We train EBU and baselines agents for 10 million frames on 49 Atari games with the same network structure, hyperparameters and evaluation methods used by Nature DQN. We divide the training steps into

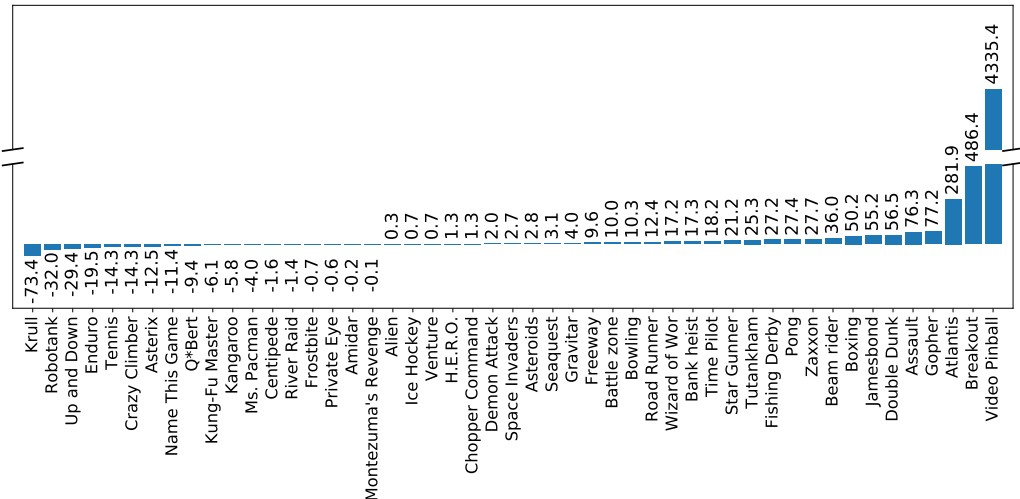

Figure 4: Relative performance (Eq.(3)) of EBU over Nature DQN in percents (%) after 10 million frames of training.

40 epochs of 250,000 frames. At the end of each epoch, we evaluate the agent for 30 episodes using $\epsilon$-greedy policy with $\epsilon = 0.05$. Transitions of the Arcade Learning Environment are fully deterministic. In order to give diversity in experience, both train and test episodes start with at most 30 no-op actions. We train each game for 8 times with different random seeds. For each agent with a different random seed, the best evaluation score during training is taken as its result. Then we report the mean score of the 8 agents as the result of the game. Detailed specifications for each baseline are described in Appendix C.

We observe that the choice of $\beta = 1$ degrades the performance in most of the games. Instead, we use $\beta = \frac{1}{2}$, which shows the best performance among $\{\frac{1}{3}, \frac{1}{2}, \frac{2}{3}, \frac{3}{4}, 1\}$ that we tried. Further fine tuning of $\beta$ may lead to a better result.

First, we show the improvements of EBU over Nature DQN for all 49 games in Figure 4. To compare the performance of an agent over its baseline, we use the following measure (Wang et al., 2015).

$$\frac{\text{Score}_{\text{Agent}} - \text{Score}_{\text{Baseline}}}{\max\{\text{Score}_{\text{Human}}, \text{Score}_{\text{Baseline}}\} - \text{Score}_{\text{Random}}}. \tag{3}$$

This measure shows how well the agent performs the task compared to its level of difficulty. Out of the 49 games, our agent shows better performance in 32 games. Not only that, for games such as 'Atlantis', 'Breakout' and 'Video Pinball', our agent shows significant amount of improvements. In order to compare the overall performance of an algorithm, we use Eq.(4) to calculate the human normalized score (van Hasselt et al., 2015).

$$\frac{\text{Score}_{\text{Agent}} - \text{Score}_{\text{Random}}}{|\text{Score}_{\text{Human}} - \text{Score}_{\text{Random}}|}. \tag{4}$$

We report the mean and median of the human normalized scores of 49 games in Table 2. The result shows that our algorithm outperforms the baselines in both mean and median of the human normalized scores. Furthermore, our method requires only about 37% of computation time used by Optimality Tightening[1]. Since Optimality Tightening has to calculate the Q-values of neighboring states and compare them to generate the penalty term, it requires about 3 times more training time than Nature DQN. Since EBU performs iterative episodic updates using the temporary Q-table that is shared by all transitions in the episode, its computational cost is almost the same as that of Nature DQN. Scores for each game after 10 million frames of training are summarized in Appendix A.

We show the performance of EBU and the baselines for 4 games 'Assault', 'Breakout', 'Gopher' and 'Video Pinball' in Figure 5. EBU with a diffusion coefficient $\beta = 0.5$ shows competitive

---

[1]We used NVIDIA Titan X for all experiments.

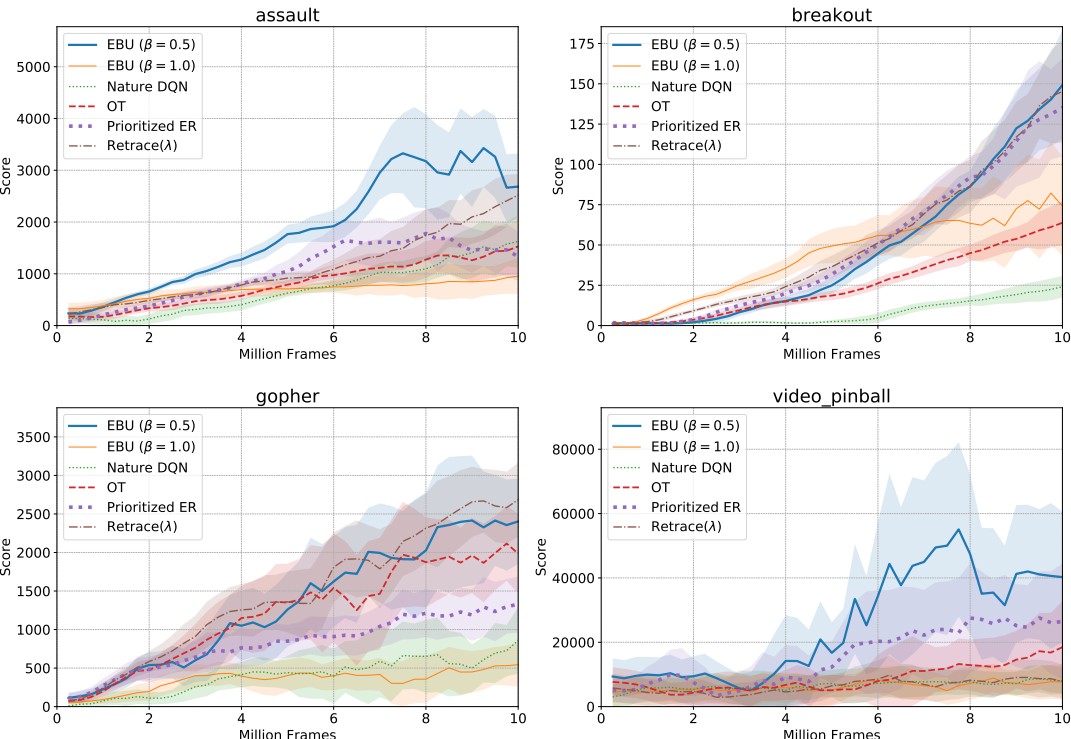

Figure 5: Scores of EBU and baselines on 4 games: 'Assault', 'Breakout', 'Gopher' and 'Video Pinball'. Moving average test scores of 40 epochs with window size 4 are plotted. The filled regions indicate the standard deviation of results from 8 random seeds.

performances in all 4 games, reflecting that our algorithm does not suffer from the stability issue caused by the sequential update of correlated states. Other baselines fail in some games, whereas our algorithm shows stable learning processes throughout all games. Out of 49 games, our algorithm shows the worst performance in only 6 games. Such stability leads to the best median and mean scores in total. Note that naive backward algorithm with $\beta = 1.0$ fails in most games.

Table 2: Summary of training time and human normalized performance. Training time refers to the total time required to train 49 games of 10M frames each (490M frames in total).

|  | Training Time (49 games, 1 GPU) | Mean | Median |
|---|---|---|---|
| EBU (10M) | 152 hours | **255.19%** | **53.65%** |
| Nature DQN (10M) | **138 hours** | 133.95% | 40.42% |
| Optimality Tightening (10M) | 407 hours | 162.66% | 49.42% |
| Prioritized Experience Replay (10M) | 146 hours | 156.57% | 40.86% |
| Retrace($\lambda$) (10M) | 154 hours | 93.77% | 41.99% |

## 5 CONCLUSION

We propose Episodic Backward Update, which samples transitions episode by episode and updates values recursively in a backward manner. Our algorithm achieves fast and stable learning due to the efficient value propagation. We show that our algorithm outperforms other baselines in many complex domains without much increase in computational cost. Since we did not change any network structures, hyperparameters and exploration methods, we hope that there is plenty of room left for further improvement.

ACKNOWLEDGEMENTS

We used the code[2] uploaded by the authors of Optimality Tightening (He et al., 2017) to evaluate the baseline. We implemented our EBU algorithm by modifying the code.

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

APPENDIX A    SCORES ACROSS ALL 49 GAMES

Table 3: Summary of raw scores after 10 million frames of trainig. Mean scores from 8 random seeds are used.

|  | EBU | Nature DQN | OT | Retrace | Prioritized ER |
|---|---|---|---|---|---|
| Alien | 708.08 | 690.32 | **1078.67** | 708.29 | 1026.96 |
| Amidar | 122.07 | 125.42 | **220.00** | 182.68 | 167.63 |
| Assault | **4109.18** | 2426.94 | 2499.23 | 2989.05 | 2720.69 |
| Asterix | 1898.12 | **2936.54** | 2592.50 | 1798.54 | 2218.54 |
| Asteroids | **1002.17** | 654.99 | 985.88 | 886.92 | 993.50 |
| Atlantis | 66271.67 | 20666.84 | 57520.00 | **98182.81** | 35665.83 |
| Bank heist | 359.62 | 234.70 | **407.42** | 223.50 | 312.96 |
| Battle zone | 26002.44 | 22468.75 | 20400.48 | **30128.36** | 20835.74 |
| Beam rider | 5628.99 | 3682.92 | **5889.54** | 4093.76 | 4586.07 |
| Bowling | **78.80** | 65.23 | 53.45 | 42.62 | 42.74 |
| Boxing | 55.95 | 37.28 | **60.89** | 6.76 | 4.64 |
| Breakout | **174.76** | 28.36 | 75.00 | 171.86 | 164.22 |
| Centipede | 6052.38 | **6207.30** | 5277.79 | 5986.16 | 4385.41 |
| Chopper Command | 1287.08 | 1168.67 | **1615.00** | 1353.76 | 1344.24 |
| Crazy Climber | 65329.63 | 74410.74 | **92972.08** | 64598.21 | 53166.47 |
| Demon Attack | **7924.14** | 7772.39 | 6872.04 | 6450.84 | 4446.03 |
| Double Dunk | -16.19 | -17.94 | -15.92 | -15.81 | **-15.62** |
| Enduro | 415.59 | 516.10 | **615.05** | 208.10 | 308.75 |
| Fishing Derby | **-39.13** | -65.53 | -69.66 | -75.74 | -78.49 |
| Freeway | **19.07** | 16.24 | 14.63 | 15.26 | 9.35 |
| Frostbite | 437.92 | 466.02 | **2452.75** | 825.00 | 536.00 |
| Gopher | 3318.50 | 1726.52 | 2869.08 | **3410.75** | 1833.67 |
| Gravitar | 294.58 | 193.55 | 263.54 | 272.08 | **319.79** |
| H.E.R.O. | 3089.90 | 2767.97 | **10698.25** | 3079.43 | 3052.04 |
| Ice Hockey | **-4.71** | -4.79 | -5.79 | -6.13 | -7.73 |
| Jamesbond | 391.67 | 183.35 | 325.21 | **436.25** | 421.46 |
| Kangaroo | 535.83 | 709.88 | 708.33 | 538.33 | **782.50** |
| Krull | 7587.24 | **24109.14** | 7468.70 | 6346.40 | 6642.58 |
| Kung-Fu Master | 20578.33 | 21951.72 | **22211.25** | 18815.83 | 18212.89 |
| Montezuma's Revenge | 1.04 | **3.95** | 0.00 | 0.00 | 0.43 |
| Ms. Pacman | 1249.79 | **1861.80** | 1849.00 | 1310.62 | 1784.75 |
| Name This Game | 6960.46 | **7560.33** | 7358.25 | 6094.08 | 5757.03 |
| Pong | 5.53 | -2.68 | 2.60 | 8.65 | **12.83** |
| Private Eye | 953.58 | **1388.45** | 1277.53 | 714.97 | 269.28 |
| Q*Bert | 785.00 | 2037.21 | **3955.10** | 3192.08 | 1215.42 |
| River Raid | 3460.62 | 3636.72 | 4643.62 | 4178.92 | **6005.62** |
| Road Runner | 10086.74 | 8978.17 | **19081.55** | 9390.83 | 17137.92 |
| Robotank | 11.65 | **16.11** | 12.17 | 9.90 | 6.46 |
| Seaquest | 1380.67 | 762.10 | 2170.33 | **2275.83** | 1955.67 |
| Space Invaders | 797.29 | 755.95 | **869.83** | 783.35 | 762.54 |
| Star Gunner | 2737.08 | 708.66 | 1710.83 | **2856.67** | 2629.17 |
| Tennis | -3.41 | **0.00** | -6.37 | -2.50 | -10.32 |
| Time Pilot | 3505.42 | 3076.98 | 4012.50 | 3651.25 | **4434.17** |
| Tutankham | 204.83 | 165.27 | 247.81 | 156.16 | **255.74** |
| Up and Down | 6841.83 | **9468.04** | 6706.83 | 7574.53 | 7397.29 |
| Venture | 105.10 | 96.70 | **106.67** | 50.85 | 60.40 |
| Video Pinball | **84859.24** | 17803.69 | 38528.58 | 18346.58 | 55646.66 |
| Wizard of Wor | **1249.89** | 529.85 | 1177.08 | 1083.69 | 1175.24 |
| Zaxxon | 3221.67 | 685.84 | 2467.92 | 596.67 | **3928.33** |

## APPENDIX B    AVERAGE PERFORMANCE OVER ALL 49 GAMES

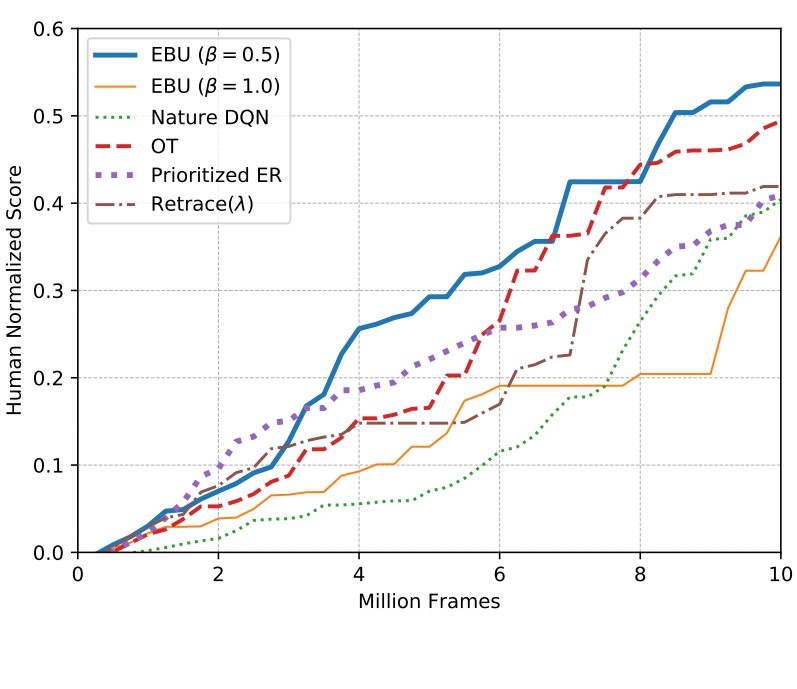

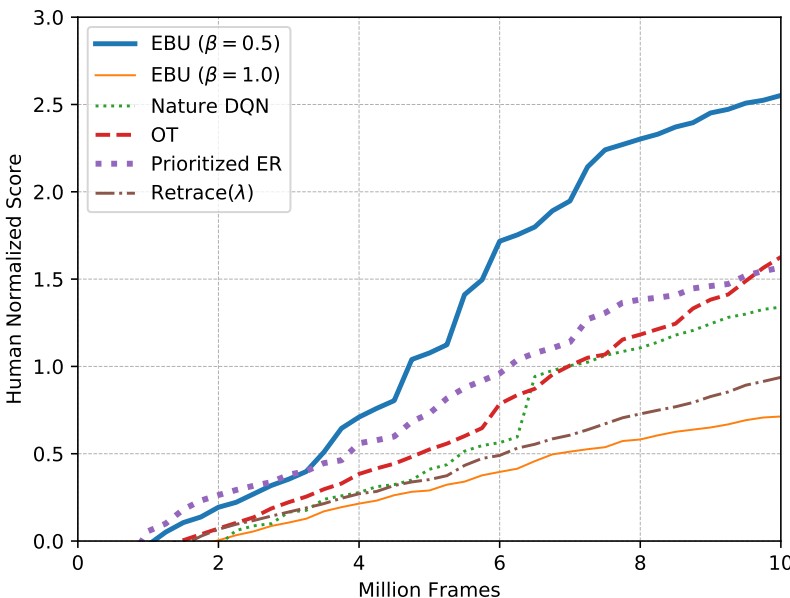

Figure 6: Learning curves of EBU and baselines on 49 games of the Arcade Learning Environment. **Up**: median over 49 games. **Down**: mean over 49 games. We use the average of 30 no-op scores of 8 agents with different random seeds. Use of different evaluation methods and seeds may output different results.

APPENDIX C    NETWORK STRUCTURE AND HYPERPARAMETERS

**2D MNIST MAZE ENVIRONMENT**

Each state is given as a grey scale $28 \times 28$ image. We apply 2 convolutional neural networks (CNNs) and one fully connected layer to get the output Q- values for 4 actions: up, down, left and right. The first CNN uses 64 channels with $4 \times 4$ kernels and stride of 3. The next CNN uses 64 channels with $3 \times 3$ kernels and stride of 1. Then the layer is fully connected into size of 512. Then we fully connect the layer into size of the action space 4. After each layer, we apply rectified linear unit.

We train the agent for a total of 200,000 steps. The agent performs $\epsilon$-greedy exploration. $\epsilon$ starts from 1 and is annealed to 0 at 200,000 steps in a quadratic manner: $\epsilon = \frac{1}{(200,000)^2}(\text{step} - 200,000)^2$. We use RMSProp optimizer with a learning rate of 0.001. The online-network is updated every 50 steps, the target network is updated every 2000 steps. The replay memory size is 30000 and we use minibatch size of 350. We use a discount factor $\gamma = 0.9$ and a diffusion coefficient $\beta = 1.0$. The agent plays the game until it reaches the goal or it stays in the maze for more than 1000 time steps.

**ARCADE LEARNING ENVIRONMENT**

**Common Specifications**
Almost all specifications such as hyperparameters and network structures are identical for all baselines. We use exactly the same network structure and hyperparameters of Nature DQN (Mnih et al., 2015). The raw observation is preprocessed into gray scale image of $84 \times 84$. Then it passes through three convolutional layers: 32 channels with $8 \times 8$ kernels with stride of 4; 64 channels with $4 \times 4$ kernels with stride of 2; 64 channels with $3 \times 3$ kernels with stride of 1. Then it is fully connected into size of 512. Then it is again fully connected into the size of the action space.

We train agents for 10 million frames each, which is equivalent to 2.5 million steps with frame skip of 4. The agent performs $\epsilon$-greedy exploration. $\epsilon$ starts from 1 and is linearly annealed to reach the final value 0.1 at 4 million frames of training. To give randomness in experience, we select a number k from 1 to 30 uniform randomly at the start of each train and test episode. We start the episode with k no-op actions. The network is trained by RMSProp optimizer with a learning rate of 0.00025. At each step, we update transitions in minibatch with size 32. The replay buffer size is 1 million steps (4 million frames). The target network is updated every 10,000 steps. The discount factor is $\gamma = 0.99$.

We divide the training process into 40 epochs of 250,000 frames each. At the end of each epoch, the agent is tested for 30 episodes with $\epsilon = 0.05$. The agent plays the game until it runs out of lives or time (18,000 frames, 5 minutes in real time).

Below are detailed specifications for each algorithm.

**1. Episodic Backward Update**
We set the diffusion coefficient $\beta = 0.5$.

**2. Optimality Tightening**
To generate the lower and upper bounds, we use 4 future transitions and 4 past transitions.

**3. Prioritized Experience Replay**
As described in the paper (Schaul et al., 2016), we use the rank-based DQN version with hyperparameters $\alpha = 0.5$, $\beta = 0$.

**4. Retrace($\lambda$)**
Just as EBU, we sample a random episode and then generate the Retrace target for the transitions in the sampled episode. First, we calculate the trace coefficients from $s = 1$ to $s = T$ (terminal).

$$c_s = \lambda \min \left( 1, \frac{\pi(a_s | x_s)}{\mu(a_s | x_s)} \right).$$ (5)

Where $\mu$ is the behavior policy of the sampled transition and the evaluation policy $\pi$ is the current policy. Then we generate a loss vector for transitions in the sample episode from $t = T$ to $t = 1$.

$$\Delta Q(x_{t-1}, a_{t-1}) = c_t \lambda \Delta Q(x_t, a_t) + [r(x_{t-1}, a_{t-1}) + \gamma \mathbb{E}_\pi Q(x_t, :) - Q(x_{t-1}, a_{t-1})].$$ (6)

## APPENDIX D SUPPLEMENTARY FIGURE: BACKWARD UPDATE ALGORITHM

Line #10 of Algorithm 2: Sample a random episode $E$.

Line # 11~13: Generate a temporary target Q table $\tilde{Q}$ with the next state vector $S'$. Initialize a target vector $y$.
Let there be $n$ possible actions in the environment. $\mathcal{A} = \{a^{(1)}, a^{(2)}, ..., a^{(n)}\}$.
Note that $\hat{Q}$ is the target Q-value and $\hat{Q}(S_{T+1}, :) = 0$.

Line # 14~17, first iteration (k = T-1): Update $\tilde{Q}$ and $y$. Let the T-th action in the replay memory be $A_T = a^{(2)}$.
① line # 15: update $\tilde{Q}[A_{k+1}, k] = \tilde{Q}[A_T, T-1] = \tilde{Q}[a^{(2)}, T-1] \leftarrow \beta\, y_T + (1-\beta)\hat{Q}(S_T, a^{(2)})$
② line # 16: update $y_k = y_{T-1} \leftarrow R_{T-1} + \gamma \max \tilde{Q}[:, T-1]$

Line # 14~17, second iteration (k = T-2): Update $\tilde{Q}$ and $y$. Let the (T-1)-th action in the replay memory be $A_{T-1} = a^{(1)}$.
① line # 15: update $\tilde{Q}[A_{k+1}, k] = \tilde{Q}[A_{T-1}, T-2] = \tilde{Q}[a^{(1)}, T-2] \leftarrow \beta\, y_{T-1} + (1-\beta)\hat{Q}(S_{T-1}, a^{(1)})$
② line # 16: update $y_k = y_{T-2} \leftarrow R_{T-2} + \gamma \max \tilde{Q}[:, T-2]$

Repeat this update until k =1.

Figure 7: Target generation process from the sampled episode E

APPENDIX E    THEORETICAL GUARANTEES

Now, we will prove that the episodic backward update algorithm converges to the true action-value function $Q^*$ in the case of finite and deterministic environment.

**Definition 1.** *(Deterministic MDP)*

*$M = (S, A, P, R)$ is a **deterministic MDP** if $\exists g : S \times A \to S$ s.t.*

$$P(s'|s, a) = \begin{cases} 1 & if \ s' = g(s, a) \\ 0 & else \end{cases} \forall (s, a, s') \in S \times A \times S,$$

In the episodic backward update algorithm, a single (state, action) pair can be updated through multiple episodes, where the evaluated targets of each episode can be different from each other. Therefore, unlike the bellman operator, episodic backward operator depends on the exploration policy for the MDP. Therefore, instead of expressing different policies in each state, we define a schedule to represent the frequency of every distinct episode (which terminates or continues indefinitely) starting from the target (state, action) pair.

**Definition 2.** *(Schedule)*

*Assume a MDP $M = (S, A, P, R)$ , where $R$ is a bounded function. Then, for each state $(s, a) \in S \times A$ and $j \in [1, \infty]$, we define $j$-**length path set** $p_{s,a}(j)$ and **path set** $p(s, a)$ for $(s, a)$ as*

$$p_{s,a}(j) = \left\{ (s_i, a_i)_{i=0}^{j} | (s_0, a_0) = (s, a), P(s_{i+1}|s_i, a_i) > 0 \quad \forall i \in [0, j-1], s_j \quad is \quad terminal \right\}.$$

*and $p_{s,a} = \cup_{j=1}^{\infty} p_{s,a}(j)$.*

*Also, we define a **schedule set** $\lambda_{s,a}$ for (state action) pair $(s, a)$ as*

$$\lambda_{s,a} = \left\{ (\lambda_i)_{i=1}^{|p_{s,a}|} | \sum_{i=1}^{|p_{s,a}|} \lambda_i = 1, \lambda_i > 0 \quad \forall i \in [1, |p_{s,a}|] \right\}.$$

*Finally, to express the varying schedule in time at the RL scenario, we define a **time schedule set** $\lambda$ for MDP $M$ as*

$$\lambda = \left\{ \{\lambda_{s,a}(t)\}_{(s,a) \in S \times A, t=1}^{\infty} | \lambda_{s,a}(t) \in \lambda_{s,a} \forall (s, a) \in S \times A, t \in [1, \infty] \right\}.$$

Since no element of the path can be the prefix of the others, the path set corresponds to the enumeration of all possible episodes starting from each (state, action) pair. Therefore, if we utilize multiple episodes from any given policy, we can see the empirical frequency for each path in the path set belongs to the schedule set. Finally, since the exploration policy can vary across time, we can group independent schedules into the time schedule set.

For a given time schedule and MDP, now we define the episodic backward operator.

**Definition 3.** *(Episodic backward operator)*

*Assume a MDP $M = (S, A, P, R)$, and time schedule $\{\lambda_{s,a}(t)\}_{t=1,(s,a) \in S \times A}^{\infty} \in \lambda$.*

*Then, **episodic backward operator** $H_t^{\beta}$ is defined as*

$$(H_t^{\beta} Q)(s, a) \tag{7}$$

$$= \mathbb{E}_{s' \in S, P(s'|s,a)} \left[ r(s, a, s') + \gamma \sum_{i=1}^{|p_{s,a}|} (\lambda_{(s,a)}(t))_i \mathbb{1}(s_{i1} = s') \left[ \max_{1 \le j \le |(p_{s,a})_i|} T_{(p_{s,a})_i}^{\beta,Q}(j) \right] \right].$$

$$T^{\beta,Q}_{(p_{s,a})_i}(j) \tag{8}$$

$$= \sum_{k=1}^{j-1} \beta^{k-1}\gamma^{k-1} \left\{ \beta r(s_{ik}, a_{ik}, s_{i(k+1)}) + (1-\beta)Q(s_{ik}, a_{ik}) \right\} + \beta^{j-1}\gamma^{j-1} \max_{a \neq a_j} Q(s_{ij}, a_{ij}).$$

*Where $(p_{s,a})_i$ is the $i$-th path of the path set, and $(s_{ij}, a_{ij})$ corresponds to the $j$-th (state, action) pair of the $i$-th path.*

Episodic backward operator consists of two parts. First, given the path that initiates from the target (state, action) pair, function $T^{\beta,Q}_{(p_{s,a})_i}$ computes the maximum return of the path via backward update. Then, the return is averaged by every path in the path set. Now, if the MDP $M$ is deterministic, we can prove that the episodic backward operator is a contraction in the sup-norm, and the fixed point of the episodic backward operator is the optimal action-value function of the MDP regardless of the time schedule.

**Theorem 2.** *(Contraction of episodic backward operator and the fixed point)*

*Suppose $M = (S, A, P, R)$ is a deterministic MDP. Then, for any time schedule $\{\lambda_{s,a}(t)\}_{t=1,(s,a) \in S \times A}^{\infty} \in \lambda$, $H_t^\beta$ is a contraction in the sup-norm for any $t$, i.e*

$$\|(H_t^\beta Q_1) - (H_t^\beta Q_2)\|_\infty \leq \gamma \|Q_1 - Q_2\|_\infty. \tag{9}$$

*Furthermore, for any time schedule $\{\lambda_{s,a}(t)\}_{t=1,(s,a) \in S \times A}^{\infty} \in \lambda$, the fixed point of $H_t^\beta$ is the optimal $Q$ function $Q^*$.*

*Proof.* First, we prove $T^{\beta,Q}_{(p_{s,a})_i}(j)$ is a contraction in the sup-norm for all $j$.

Since $M$ is a deterministic MDP, we can reduce the return as

$$T^{\beta,Q}_{(p_{s,a})_i}(j) = \left( \sum_{k=1}^{j-1} \beta^{k-1}\gamma^{k-1} \left\{ \beta r(s_{ik}, a_{ik}) + (1-\beta)Q(s_{ik}, a_{ik}) \right\} + \beta^{j-1}\gamma^{j-1} \max_{a \neq a_j} Q(s_{ij}, a_{ij}) \right). \tag{10}$$

$$\|T^{\beta,Q_1}_{(p_{s,a})_i}(j) - T^{\beta,Q_2}_{(p_{s,a})_i}(j)\|_\infty \leq \left\{ (1-\beta) \sum_{k=1}^{j-1} \beta^{k-1}\gamma^{k-1} + \beta^{j-1}\gamma^{j-1} \right\} \|Q_1 - Q_2\|_\infty$$

$$= \left\{ \frac{(1-\beta)(1-(\beta\gamma)^{j-1})}{1-\beta\gamma} + \beta^{j-1}\gamma^{j-1} \right\} \|Q_1 - Q_2\|_\infty$$

$$= \frac{1 - \beta + \beta^j\gamma^{j-1} - \beta^j\gamma^j}{1-\beta\gamma} \|Q_1 - Q_2\|_\infty$$

$$= \left\{ 1 + (1-\gamma)\frac{\beta^j\gamma^{j-1} - \beta}{1-\beta\gamma} \right\} \|Q_1 - Q_2\|_\infty$$

$$\leq \|Q_1 - Q_2\|_\infty (\because \beta \in [0,1], \gamma \in [0,1)). \tag{11}$$

Also, at the deterministic MDP, the episodic backward operator can be reduced to

$$(H_t^\beta Q)(s,a) = r(s,a) + \gamma \sum_{i=1}^{|p_{s,a}|} (\lambda_{(s,a)})_i(t) \left[ \max_{1 \le j \le |(p_{s,a})_i|} T_{(p_{s,a})_i}^{\beta,Q}(j) \right]. \tag{12}$$

Therefore, we can finally conclude that

$$\|(H_t^\beta Q_1) - (H_t^\beta Q_2)\|_\infty$$

$$= \max_{s,a} \left| H_t^\beta Q_1(s,a) - H_t^\beta Q_2(s,a) \right|$$

$$\le \gamma \max_{s,a} \left[ \sum_{i=1}^{|p_{s,a}|} (\lambda_{(s,a)}(t))_i \left| \left\{ \max_{1 \le j \le |(p_{s,a})_i|} T_{(p_{s,a})_i}^{\beta,Q_1}(j) \right\} - \left\{ \max_{1 \le j \le |(p_{s,a})_i|} T_{(p_{s,a})_i}^{\beta,Q_2}(j) \right\} \right| \right]$$

$$\le \gamma \max_{s,a} \left[ \sum_{i=1}^{|p_{s,a}|} (\lambda_{(s,a)}(t))_i \max_{1 \le j \le |(p_{s,a})_i|} \left\{ \left| T_{(p_{s,a})_i}^{\beta,Q_1}(j) - T_{(p_{s,a})_i}^{\beta,Q_2}(j) \right| \right\} \right]$$

$$\le \gamma \max_{s,a} \left[ \sum_{i=1}^{|p_{s,a}|} (\lambda_{(s,a)}(t))_i \|Q_1 - Q_2\|_\infty \right]$$

$$= \gamma \max_{s,a} \left[ \|Q_1 - Q_2\|_\infty \right]$$

$$= \gamma \|Q_1 - Q_2\|_\infty. \tag{13}$$

Therefore, we proved that episodic backward operator is a contraction independent of the schedule. Finally, we prove that the distinct episodic backward operators in terms of schedule has same fixed point, $Q^*$. A sufficient condition to prove this is given by

$$\left[ \max_{1 \le j \le |(p_{s,a})_i|} T_{(p_{s,a})_i}^{\beta,Q^*}(j) \right] = \frac{Q^*(s,a) - r(s,a)}{\gamma} \ \forall 1 \le i \le |p_{s,a}|.$$

We will prove this by contradiction. Assume $\exists i$ s.t. $\left[ \max_{1 \le j \le |(p_{s,a})_i|} T_{(p_{s,a})_i}^{\beta,Q^*}(j) \right] \ne \frac{Q^*(s,a) - r(s,a)}{\gamma}$.

First, by the definition of $Q^*$ fuction, we can bound $Q^*(s_{ik}, a_{ik})$ and $Q^*(s_{ik}, :)$ for every $k \ge 1$ as follows.

$$Q^*(s_{ik}, a) \le \gamma^{-k} Q^*(s,a) - \sum_{m=0}^{k-1} \gamma^{m-k} r(s_{im}, a_{im}). \tag{14}$$

Note that the equality holds if and only if the path $(s_i, a_i)_{i=0}^{k-1}$ is the optimal path among the ones that start from $(s_0, a_0)$. Therefore, $\forall 1 \le j \le |(p_{s,a})_i|$, we can bound $T_{(p_{s,a})_i}^{\beta,Q^*}(j)$.

$$T^{\beta,Q}_{(p_{s,a})_i}(j)$$

$$= \sum_{k=1}^{j-1} \beta^{k-1}\gamma^{k-1}\left\{\beta r(s_{ik}, a_{ik}) + (1-\beta)Q(s_{ik}, a_{ik})\right\} + \beta^{j-1}\gamma^{j-1} max_{a\neq a_j}Q(s_{ij}, a_{ij})$$

$$\leq \left\{(\sum_{k=1}^{j-1}(1-\beta)\beta^{k-1}) + \beta^{j-1}\right\}\gamma^{-1}Q^*(s,a)$$
$$+ \sum_{k=1}^{j-1}\left\{\beta^{k-1}\gamma^{k-1}\left(\beta r(s_{ik}, a_{ik}) - \sum_{m=0}^{k-1}(1-\beta)\gamma^{m-k}r(s_{im}, a_{im})\right)\right\}$$
$$- \sum_{m=0}^{j-1}\beta^{j-1}\gamma^{j-1}\gamma^{m-j}r(s_{im}, a_{im})$$

$$= \gamma^{-1}Q^*(s,a) + \sum_{k=1}^{j-1}\beta^k\gamma^{k-1}r(s_{ik}, a_{ik})$$
$$- \sum_{m=0}^{j-2}\left\{\sum_{k=m+1}^{j-1}(1-\beta)\beta^{k-1}\gamma^{m-1}r(s_{im}, a_{im})\right\} - \sum_{m=0}^{j-1}\beta^{j-1}\gamma^{m-1}r(s_{im}, a_{im})$$

$$= \gamma^{-1}Q^*(s,a) + \sum_{m=1}^{j-1}\beta^m\gamma^{m-1}r(s_{im}, a_{im})$$
$$- \sum_{m=0}^{j-2}(\beta^m - \beta^{j-1})\gamma^{m-1}r(s_{im}, a_{im}) - \sum_{m=0}^{j-1}\beta^{j-1}\gamma^{m-1}r(s_{im}, a_{im})$$

$$= \gamma^{-1}Q^*(s,a) - \gamma^{-1}r(s_{i0}, a_{i0}) = \frac{Q^*(s,a) - r(s,a)}{\gamma}.$$

$$(15)$$

Since this occurs for any arbitrary path, the only remaining case is when

$\exists i$ s.t. $\left[max_{1\leq j\leq |(p_{s,a})_i|} T^{\beta,Q^*}_{(p_{s,a})_i}(j)\right] < \frac{Q^*(s,a)-r(s,a)}{\gamma}.$

Now, let's speculate on the path $s_0, s_1, s_2, ...., s_{|(p_{s,a})_i)|}$. Let's first prove the contradiction when the length of the contradictory path is finite. If $Q^*(s_{i1}, a_{i1}) < \gamma^{-1}(Q^*(s,a) - r(s,a))$, then by the bellman equation, there exists action $a \neq a_{i1}$ s.t $Q^*(s_{i1}, a) = \gamma^{-1}(Q^*(s,a) - r(s,a))$. Then, we can find that $T^{\beta,Q^*}_{(p_{s,a})_1}(1) = \gamma^{-1}(Q^*(s,a) - r(s,a))$ so it contradicts the assumption. Therefore, $a_{i1}$ should be the optimal action in $s_{i1}$.

Repeating the procedure, we can find that $a_{i1}, a_{i2}, ..., a_{|(p_{s,a})_i)|-1}$ are optimal with respect to their corresponding states.

Finally, we can find that $T^{\beta,Q^*}_{(p_{s,a})_1}(|(p_{s,a})_i)|) = \gamma^{-1}(Q^*(s,a) - r(s,a))$ since all the actions satisfies the optimality condition of the inequality in equation 7. Therefore, it is a contradiction to the assumption.

In the case of infinite path, we will prove that for any $\epsilon > 0$, there is no path that satisfy $\frac{Q^*(s,a)-r(s,a)}{\gamma} - \left[max_{1\leq j\leq |(p_{s,a})_i|} T^{\beta,Q^*}_{(p_{s,a})_i}(j)\right] = \epsilon.$

Since the reward function is bounded, we can define $r_{\max}$ as the supremum norm of the reward function. Define $q_{\max} = \max_{s,a} |Q(s,a)|$ and $R_{\max} = \max\{r_{\max}, q_{\max}\}$. We can assume $R_{\max} > 0$. Then, let's set $n_\epsilon = \lceil \log_\gamma \frac{\epsilon(1-\gamma)}{R_{\max}} \rceil + 1$. Since $\gamma \in [0,1)$, $R_{\max} \frac{\gamma^{n_\epsilon}}{1-\gamma} < \epsilon$. Therefore, by applying the procedure on the finite path case for $1 \le j \le n_\epsilon$, we can conclude that the assumption leads to a contradiction. Since the previous $n_\epsilon$ trajectories are optimal, the rest trajectories can only generate a return less than $\epsilon$.

Finally, we proved that $\left[ \max_{1 \le j \le |(p_{s,a})_i|} T^{\beta,Q^*}_{(p_{s,a})_i}(j) \right] = \frac{Q^*(s,a) - r(s,a)}{\gamma}$ $\forall 1 \le i \le |p_{s,a}|$ and therefore, every episodic backward operator has $Q^*$ as the fixed point. $\qquad\square$

Finally, we will show that the online episodic backward update algorithm converges to the optimal $Q$ function $Q^*$.

**Restatement of Theorem 1.** *Given a finite, deterministic, and tabular MDP $M = (S, A, P, R)$, the episodic backward update algorithm, given by the update rule*

$$Q_{t+1}(s_t, a_t)$$

$$= (1 - \alpha_t)Q_t(s_t, a_t) + \alpha_t \left[ r(s_t, a_t) + \gamma \sum_{i=1}^{|p_{s_t, a_t}|} (\lambda_{(s_t, a_t)})_i(t) \left[ \max_{1 \le j \le |(p_{s_t, a_t})_i|} T^{\beta,Q}_{(p_{s_t, a_t})_i}(j) \right] \right]$$

*converges to the optimal $Q$ function w.p. 1 as long as*

- *The step size satisfies the Robbins-Monro condition;*

- *The sample trajectories are finite in lengths $l$: $\mathbb{E}[l] < \infty$;*

- *Every (state, action) pair is visited infinitely often.*

For the proof of Theorem 1, we follow the proof of Melo, 2001.

**Lemma 1.** *The random process $\Delta_t$ taking values in $\mathbb{R}^n$ and defined as*

$$\Delta_{t+1}(x) = (1 - \alpha_t(x))\Delta_t(x) + \alpha_t(x)F_t(x)$$

*converges to zero w.p.1 under the following assumptions:*

- $0 \le \alpha_t \le 1, \sum_t \alpha_t(x) = \infty$ *and* $\sum_t \alpha_t^2(x) < \infty$;

- $\|\mathbb{E}[F_t(x)|\mathcal{F}_t]\|_W \le \gamma\|\Delta_t\|_W$, *with* $\gamma < 1$;

- $\mathbf{var}[F_t(x)|\mathcal{F}_t] \le C\left(1 + \|\Delta_t\|_W^2\right)$, *for* $C > 0$.

By Lemma 1, we can prove that the online episodic backward update algorithm converges to the optimal $Q^*$.

*Proof.* First, by assumption, the first condition of Lemma 1 is satisfied. Also, we can see that by substituting $\Delta_t(s,a) = Q_t(s,a) - Q^*(s,a)$, and $F_t(s,a) = r(s,a) + \gamma \sum_{i=1}^{|p_{s,a}|} (\lambda_{(s,a)})_i(t) \left[ \max_{1 \le j \le |(p_{s,a})_i|} T^{\beta,Q}_{(p_{s,a})_i}(j) \right] - Q^*(s,a)$. $\|\mathbb{E}[F_t(s,a)|\mathcal{F}_t]\|_\infty = \|(H_t^\beta Q_t)(s,a) - (H_t^\beta Q^*)(s,a)\|_\infty \le \gamma\|\Delta_t\|_\infty$, where the inequality holds due to the contraction of the episodic backward operator.

Then, $\mathbf{var}[F_t(x)|\mathcal{F}_t] = \mathbf{var}\left[ r(s,a) + \gamma \sum_{i=1}^{|p_{s,a}|} (\lambda_{(s,a)})_i(t) \left[ \max_{1 \le j \le |(p_{s,a})_i|} T^{\beta,Q}_{(p_{s,a})_i}(j) \right] \middle| \mathcal{F}_t \right]$.

Since the reward function is bounded, the third condition also holds as well. Finally, by Lemma 1, $Q_t$ converges to $Q^*$.

$\qquad\square$

Although the episodic backward operator can accommodate infinite paths, the operator can be practical when the maximum length of the episode is finite. This assumption holds for many RL domains, such as ALE.

