# OpenReview forum: "Sample-Efficient Deep Reinforcement Learning via Episodic Backward Update"
_ICLR.cc/2018/Conference — Reject_

### Official Review · AnonReviewer2 · 2017-11-27
**A potentially interesting approach, but with weak theoretical and empirical validation**

**Rating:** 4
**Confidence:** 4

**Review:**

This paper proposes a new variant of DQN where the DQN targets are computed on a full episode by a « backward » update (i.e. from end to start of episode). The targets’ update rule is similar to a regular tabular Q-learning update with high learning rate beta: this allows faster propagation of rewards obtained at the end of the episode (while beta=0 corresponds to regular DQN with no such reward propagation). This mechanism is shown to improve on Q-learning in a toy 2D maze environment (with MNIST-based pixel states providing cell coordinates) with beta=1, and on DQN and its optimality tightening variant on Atari games with beta=0.5.

The intuition behind the algorithm (that one should try to speed up the propagation of rewards across multiple steps) is not new, in fact it has inspired other approaches like n-step Q-learning, eligibility traces or more recently Retrace(lambda) in deep RL. Actually the idea of replaying experiences in backward order can be traced back to the origins of experience replay («  Programming Robots Using Reinforcement Learning and Teaching », Lin, 1991), something that is not mentioned here. That being said, to the best of my knowledge the specific algorithm proposed in this submission (Alg. 2) is novel, even if Alg. 1 is not (Alg. 1 can be seen as a specific instance of Lin’s algorithm with a very high learning rate, and clearly only makes sense in toy deterministic environments).

In the absence of any theoretical analysis of the proposed approach, I would have expected an in-depth empirical validation. Unfortunately this is not the case here. In the toy environment (4.1) I am surprised by the really poor quality of the results (paths 5-10 times longer than the shortest path on average): have algorithms been run for a long enough time? Or maybe the average is a bad performance measure due to outliers? I would have also appreciated a comparison to Retrace(lambda), which is a more principled way to use multi-step rewards than n-step Q-learning (which is technically an on-policy method). Similar remarks can be made on the Atari experiments (4.2), where 10M frames is really low (the original DQN paper had results on 50M frames, and Rainbow reports 200M frames in only ~2x the training time reported here). The comparison also should have included prioritized experience replay, which has been shown to provide a significant boost in DQN, but may be tricky to combine with the proposed algorithm. Overall comparing only to vanilla DQN and its optimality tightening variant is too limited when there have been so many other meaningful improvements over DQN. This makes it really hard to tell whether the proposed algorithm would actually help when combined with a state-of-the-art method like Rainbow for instance.

A few additional small remarks and questions:
- « Second, there is no point in updating a one-step transition unless the future transitions have not been updated yet. »: should « unless » be replaced by « if »?
- In 4.1 is there a maximum number of steps per episode and can you please confirm that training is done independently for each maze?
- Typo in eq. 3: the - in the max should be a comma
- There is a good amount of typos and grammar errors, though they do not harm the readability of the paper
- Citations for « Deep Reinforcement Learning with Double Q-learning » and « Dueling Network Architectures for Deep Reinforcement Learning » could refer to their conference versions
- « epsilon starts from 1 and is annealed to 0 at 200,000 steps in a quadratic manner »: please specify the exact formula
- Fig. 7 is really confusing, there seem to be typos and it is not clear why the beta updates appear in these specific cells, please revise it if you want to keep it

---

> ### Author Response · Authors · 2017-12-03
> **Answers to questions and our plan to revise the paper**
>
> Thank you for your detailed feedback and questions.
> I'd like to answer some of questions and share our plan to revise the paper with regards to your feedback.
>
> 1. Limited comparison
> We strongly agree that we need more baseline algorithms to show the effectiveness of our algorithm. As other reviewers have suggested, we will include the performance of prioritized experience replay and retrace algorithm in the revised version.
>
> 2.  Idea of replaying experiences in backward
> Thank you for the reference, we will mention the relationship between Lin's idea and our methods in the revised version.
>
> 3. Poor performance in MNIST DQN
> Learning curve tends to converge so fast for all algorithms when we used simple 2D maze, so it was difficult to compare different algorithms. So we used MNIST images as the state representation to make the learning process of general state transitions harder. We trained the agents for 200,000 steps, and all three algorithms (backward DQN, vanilla DQN, n-step DQN) converge to 1. In the paper, we showed the plots over 100,000 steps to show the effectiveness of our method in the early stages of training. To avoid any confusion, we will show the results until 200,000 steps in the revised version. Note that the vanilla DQN is trained for 50M steps (200M frames) in the Atari domain. Since the MNIST DQN environment is much simple, it is reasonable that the training is done for 0.2M steps.
>
> 4. A few more comments on MNIST DQN:
> We terminated the episode when the agent stays in the maze for more than 1000 time steps.
> We trained 50 different independent agents each in a different random maze and reported the mean score. But as you suggested, mean may be a bad measure due to outliers. So we will show both mean and median of 50 agents' scores as the result in the revised version.
>
> 5. Running time compared to RAINBOW
> Running time may vary a lot depending on which device and distributed method you use. We used a single GPU to train an agent. As reported in the paper, it took 152 hours to train 490M frames (49 games x 10M frames). RAINBOW takes 10 days to train 200M frames. We will mention that the training time is not the 'mean' training time of 49 games but the 'sum' of training time in the revised version.
>
> 6. The last figure
> We apologize for the confusion. The first column and fourth rows of initialization and recursive updates part should be changed as "s_1" -> "s_2". The beta is applied only for the positions where the actions were taken in the replay memory, as the update is done from right to left.  a_T = A_2, a_(T-1) = A_1 in the example. We will make this clear in the revised version.
>
> 7. Typos and Citations
> We will correct the typos and citations as your suggestions.
>
> Thank you so much for your ideas and suggestions.
> Any further comments are appreciated.

---

> > ### Comment · AnonReviewer2 · 2018-01-10
> > **Re: Answers to questions and our plan to revise the paper**
> >
> > Thanks for the reply, and for the revisions to the manuscript. It's great that you added more material, in particular more experiments and a theoretical analysis. Unfortunately I'm afraid it's a bit too much for a paper revision, as it would require a re-review, thus I am reluctant to improve my score without re-reading the whole thing carefully (which I lack time for).
> >
> > I'm still quite concerned by the small number of steps (10M) in experiments. I guess you are limited by your single GPU, which is sad, but I don't think one can draw meaningful conclusions on such small-scale experiments. I'm worried that prioritized experience replay doesn't seem to work much better than Vanilla DQN (no improvement on Median score for instance), while previous work suggests it is an important ingredient (ex: Dueling networks, Rainbow). Assuming this is not an implementation issue, the small number of steps could be the culprit.

---

### Official Review · AnonReviewer1 · 2017-11-27
**An RL update for DQN-like agents based on recursive max backups.**

**Rating:** 6
**Confidence:** 4

**Review:**

The authors propose a simple modification to the DQN algorithm they call Episodic Backward Update. The algorithm selects transitions in a backward order fashion from end of episode to be more effective in propagating learning of new rewards. This issue of fast propagation of updates is a common theme in RL (cf eligibility traces, prioritised sweeping, and more recently DQN with prioritised replay etc.). Here the proposed update applies the max Bellman operator recursively on a trajectory (unsure whether this is novel), with some decay to prevent accumulating errors with the nested max.

The paper is written in a clear way. The proposed approach seems reasonable, but I would have guessed that prioritized replay would also naturally sample transitions in roughly that order - given that TD-errors would at first be higher towards the end of an episode and progress backwards from there. I think this should have been one of the baselines to compare to for that reason.

The experimental results seem promising in the illustrative MNIST domain. Atari results seem decent, especially given that experiments are limited to 10M frames, though the advantage compared to the related approach of optimality tightening is not obvious.

---

> ### Author Response · Authors · 2017-12-03
> **Our plan to revise the paper**
>
> Thank you for your time and suggestions.
>
> As you mentioned, we guess there may be some relation between prioritized experience replay and our method. As all the reviewers have mentioned, we will add prioritized experience replay and retrace algorithm as the baseline to compare in the revised version.
>
> Any further suggestions are appreciated.

---

### Official Review · AnonReviewer3 · 2017-12-01
**The paper is interesting, but it lacks the proper comparisons to previously published techniques.**

**Rating:** 5
**Confidence:** 4

**Review:**

This paper proposes a new way of sampling data for updates in deep-Q networks. The basic principle is to update Q values starting from the end of the episode in order to facility quick propagation of rewards back along the episode.

The paper is interesting, but it lacks the proper comparisons to previously published techniques.

The results presented by this paper shows improvement over the baseline. But the Atari results is still significantly worse than the current SOTA.

In the non-tabular case, the authors have actually moved away from Q learning and defined an objective that is both on and off-policy. Some (theoretical) analysis would be nice. It is hard to judge whether the objective defined in the non-tabular defines a contraction operator at all in the tabular case.

There has been a number of highly relevant papers. Prioritized replay, for example, could have a very similar effect to proposed approach in the tabular case.

In the non-tabular case, the Retrace algorithm, tree backup, Watkin's Q learning all bear significant resemblance to the proposed method. Although the proposed algorithm is different from all 3, the authors should still have compared to at least one of them as a baseline. The Retrace algorithm specifically has also been shown to help significantly in the Atari case, and it defines a convergent update rule.

---

> ### Author Response · Authors · 2017-12-03
> **Our plan to revise the paper**
>
> Thank you for your time and suggestions.
>
> As you and other reviewers have mentioned, we strongly agree that we lack the comparisons to other related methods. We will try to compare our results and those of prioritized experience replay and retrace algorithm in the revised version. Also we will try to add some theoretical analysis to compare our algorithm to others.
>
> Any further comments and thoughts are appreciated.

---

### Public Comment · ~Tyler_Kolody1 · 2017-12-03
**Reproducibility Challenge request**

I'm taking part in the reproducibility challenge put forward by Prof. Joelle Pineau and was wondering if we could have access to your code and any other information that might be pertinent when recreating your experiment. Any information such as hyperparameter values not mentioned in the paper and library versions would be extremely useful.

Thank you

---

> ### Author Response · Authors · 2017-12-03
> **Reproducibility of our experiment**
>
> We are planning to upload our code after the revision process since we cannot reveal our identity before the final decision.
>
> But as described in the paper, our code is built upon the codes of the paper (« Learning to Play in a Day: Faster Deep Reinforcement Learning by Optimality Tightening», He et al., 2017) https://github.com/ShibiHe/Q-Optimality-Tightening
> All the hyperparameters and network structures are the same as those of above, except that we applied the final time step of 18000 frames (5 mins) for each episode.
>
> The two major differences between our code and that of Optimality Tightening are the followings.
> 1. To implement our backward target generation, we modified the "_do_training" function of "ale_agents.py".
> 2. To sample a random episode, we defined "random_batch" function in "ale_data_set.py". This function is run only after all steps of previously sampled episode are updated.
>
> Thank you.

---

> > ### Public Comment · ~Tyler_Kolody1 · 2017-12-03
> > **Thank you**
> >
> > We really appreciate the quick response and details.
> >
> > Best of luck

---

> > ### Public Comment · ~Rajat_Bhateja1 · 2017-12-11
> > **Further questions for reproducibility**
> >
> > Hi, as part of our efforts to reproduce the experiments suggested in your paper, we wanted to ask a few more questions:
> >
> > 1. When you mention the changing the final time step to 18000 frames, did you mean the parameter freeze_interval in q_network.py and did you notice degrading steps per second when you were training it.
> >
> > 2. When running a baseline for nature DQN, how many epochs did you run it for
> >
> > 3. Is there a way to test the DQN without using Optimally Tightening, and finally
> >
> > 4. Did you make any changes or tweaks to accelerate the learning to get the training times you mentioned and if you remember, which game took the least amount of time to train.
> >
> > Once again, thanks for your previous response and hoping to hear from you soon.

---

> > > ### Author Response · Authors · 2017-12-12
> > > **Reproducibility of our experiment**
> > >
> > > Thank you for your efforts.
> > >
> > > 1. What we mean by the final time step is not the parameter freeze_interval. But the maximum number of frames that each episode can last. This is a conventional way taken by all other algorithms on the Atari domain since some games may not terminate if the agent takes no significant actions. Refer to "run_episode" function of "ale_experiment.py".
> > >
> > > 2. As mentioned in the paper, we trained the agent for 40 epochs (we set 1 epoch = 62,500 steps = 250,000 frames). So that makes a total of 10M frames of training.
> > >
> > > 3. If you mean the Nature DQN, you may want to use one of the following codes:
> > >     1) The original Lua code by Deepmind ( https://sites.google.com/a/deepmind.com/dqn/)
> > >     2) Theano based Deep-Q RL code (https://github.com/spragunr/deep_q_rl)
> > >
> > > 4. We do not really have any tweaks to accelerate the learning. But the Theano version of the DQN code tends to be faster than the original Lua code. For its simplicity, "Pong" takes the least amount of training time out of the 49 games we tried.
> > >
> > > Best of luck.

---

> > > > ### Public Comment · ~Tyler_Kolody1 · 2017-12-12
> > > > **Clarification**
> > > >
> > > > For the final step size, we found the max_steps variable and were wondering if it should be set to 18000 (the number of frames), or if the variable refers to steps and it should be set to 18000/4 frames per step = 4500? Was this change made for the OT baseline, or just the EBU code?
> > > >
> > > > Regarding outputs, I apologize if i've just overlooked something obvious in the code, but how do you get the raw scores for Appendix A? I see the 30 no-ops evaluation, but am not sure where to look/what I'm looking for regarding the output.
> > > >
> > > > UPDATE: The author of the original code mentioned they wrote something separate for those evaluations: was that the case for you as well?
> > > >
> > > > Thanks as always

---

> > > > > ### Author Response · Authors · 2017-12-13
> > > > > **Clarification**
> > > > >
> > > > > 1. The max_steps variable should be 4500 steps. For a fair comparison, we set the same parameters for EBU and other baselines.
> > > > >
> > > > > 2. There are some sources of randomness in the algorithm.
> > > > >     1) epsilon greedy exploration
> > > > >     2) sampling episode
> > > > >     3) number of steps for no-ops
> > > > > To test the robustness of the algorithm, we used 8 different random seeds for the randomness. For each seed, at the end of every epoch we test the agent for 30 episodes with epsilon = 0.05. Since one epoch is 250,000 frames and we train for 1,000,000 frames in total, we have 40 test results for an agent with single random seed. Since there are oscillations in the test score, as mentioned in the paper, we take the best result out of 40 test scores as the result of the agent with that random seed. (following common practice (van Hasselt et al., 2015; Mnih et al., 2015)). Since we have 8 agents with different random seeds, we have 8 such results and we take mean of them to output the raw score.
> > > > >
> > > > > example) suppose we have 10 epochs and 2 random seeds.
> > > > >
> > > > > epoch                     |    1    2    3    4    5    6    7    8    9    10
> > > > >
> > > > > seed 1 test score  |   10  20  30  40  50  60  40  20  50  50    --> seed 1 result = 60
> > > > >
> > > > > seed 2 test score  |    5   10  20  30  50  40  30  40  50  40     --> seed 2 result = 50
> > > > >
> > > > > We output mean of the results from all random seeds: (60+50)/2 = 55 as the result

---

> > > > > > ### Public Comment · ~Tyler_Kolody1 · 2017-12-13
> > > > > > **Clarification**
> > > > > >
> > > > > > I think my confusion stems from the fact that the only test I see is one that is dictated by steps, default=125000, rather than episodes, and explicitly is commented as "runtime evaluation, not 30 no-op evaluation" and I'm wondering how to use the 30 op evaluation. I have been unable to track down any way to get scores, but instead only get the standard 3 csv files, based on the training.
> > > > > >
> > > > > > Thank you

---

> > > > > > > ### Author Response · Authors · 2017-12-13
> > > > > > > **Clarification**
> > > > > > >
> > > > > > > I guess your source of confusion is that the same number 30 is used twice.
> > > > > > >
> > > > > > > 30 no-op evaluation method does not mean that we test the agent for 30 episodes. But means that each episode starts with at most 30 no-op actions. It is already implemented in the "_init_episode" function of "ale_experiment.py". And we take the average score of 30 (just by chance, has nothing to do with the "30" in 30 no ops evaluation) episodes generated by 30 no-op evaluation method.
> > > > > > >
> > > > > > > So we modified the "run_epoch" function and "run" function of  "ale_experiment.py".  When the "testing" parameter is true, we ran the "run_episode" function 30 times and saved the average score. So STEPS_PER_TEST = 125000 is not used.
> > > > > > >
> > > > > > > Refer to page 7 of « Learning to Play in a Day: Faster Deep Reinforcement Learning by Optimality Tightening», He et al., 2017 :
> > > > > > > ""We strictly follow the evaluation procedure in (Mnih et al., 2015) which is often referred to as ‘30
> > > > > > > no-op evaluation.’ During both training and testing, at the start of the episode, the agent always
> > > > > > > performs a random number of at most 30 no-op actions. During evaluation, our agent plays each
> > > > > > > game 30 times for up to 5 minutes, and the obtained score is averaged over these 30 runs. An -
> > > > > > > greedy policy with  = 0:05 is used. Specifically, for each run, the game episode starts with at most
> > > > > > > 30 no-op steps, and ends with ‘death’ or after a maximum of 5 minute game-play, which corresponds
> > > > > > > to 18000 frames.""

---

### Public Comment · ~yacine_mahdid1 · 2017-12-16
**Attempt at Reproducibility**

As part of our efforts to reproduce the results of the original paper, we exhaustively researched about DQN, Optimality Tightening and other algorithms in the context of the original paper, and gained an understanding of the proposed EBU algorithm and setup the environment required to implement the algorithm. We attempted to use the code provided at https://github.com/ShibiHe/
Q-Optimality-Tightening, the outline in the paper and the authors’ assistance to reproduce a subset of their results.

Obstacles to reproducibility:
The major obstacles we faced when attempting to reproduce the planned subset of the results were:
Difficulty in translating the conceptual changes pointed out by the authors in their implementation and Q-Optimality Tightening (OT) codebases.
Computational costs, and inefficiencies in the original OT code which meant that even on a Titan XP, only about 90 steps a second and 30% GPU utilization could be achieved(1).
Outputs generated from the OT code were the only ones native to the code, therefore core evaluations (assessed and compared in the paper) couldn’t be run.
Lack of original EBU implementation code coupled with a lack of prior exhaustive theoretical knowledge meant that despite the modest number of changes, implementation of EBU was very challenging.

Reproduced results: We were able to establish two kinds of baselines, first being the average reward per episode over each of the 40 epochs for 3 games in the Arcade Learning Environment (Breakout, Video Pinball and Pong) and secondly we were able to reproduce and confirm the run time for the OT baseline for all the 3 aforementioned games through extrapolation, noticing signs of decay as epoch progressed for Pong, when running on GTX970, and a few spikes when running Breakout and Video Pinball on Titan XP. We did this using the original hyperparameters of the paper (which were almost the same as the OT code), as specified by the authors. Due to the significant computational cost of each run, we were not able to attempt a wide variety of alternative parameters. We were however, unable to definitively reproduce the author’s original results. The attempt to incorporate EBU was hampered by the size and complexity of the _do_training function and a lack of comments or documentation for the original OT code, which made correlating the pseudo code to the actual code difficult.



(1). The limitation of GPU occupation saturating at 30% is also mentioned in the Readme of  Q-Optimality Tightening implementation on Github.


Tyler Kolody, Yacine Mahdid and Rajat Bhateja

---

### Author Response · Authors · 2018-01-05
**Revised Contents**

We have uploaded our revised paper. Below is the list of major revisions we made.

1. Added Lin, 1991

Original idea of replaying backward is described in the introduction and the related work sections.
Clarified that alg.1 is a special case of Lin's algorithm.

2. MNIST DQN

Figure 3 plotted until 200,000 steps.
Table 1 reports both mean and median scores at 100,000 steps.
More detailed explanations on epsilon scheduling and the time step limit in Appendix C.

3. Arcade Learning Environment

Added Prioritized ER and Retrace algorithm as baselines.
Figure 5: changed the set of games from (Atlantis, Breakout, Gopher and Pong) to (Assault, Breakout, Gopher and Video Pinball) and also included results of new baselines.
Appendix A and Appendix B: results from new baselines added. Appendix A no longer includes standard deviation information due to the margin.
Appendix C contains specifications of baselines.

4. Supplementary figure of Appendix D

Changed the notations: capital 'A' means the realizations of the sampled episode, lowercase 'a' means the possible action index of the environment.
Descirbed the update step by step: the first and second iterations.

5. Theoretical Guarantees

Added a theorem in section 3 that episodic backward update with a diffusion coefficient beta in (0,1) defines a contraction operator and converges to the optimal value in finite and deterministic MDPs.
Stated the proof in Appendix E.

---

### Decision · Program_Chairs · 2018-01-29
**ICLR 2018 Conference Acceptance Decision**

**Decision:**

Reject

**Comment:**

The reviewers agree the proposed idea is relatively incremental, and the paper itself does not do an exemplary job in other areas to make up for this.